# Large mammal population trends in Comoé National Park (1958–2022): Towards understanding their asymmetric decline and recovery in West Africa's largest savanna park

Paul Scholte[1,2]*, Olivier Pays[3,4], Bertrand Chardonnet[5], Amara Ouattara[6], Djafarou Tiomoko[2,7]

**1** Deutsche Gesellschaft für Internationale Zusammenarbeit (GIZ), Addis Ababa, Ethiopia, **2** previously Deutsche Gesellschaft für Internationale Zusammenarbeit (GIZ), Abidjan, Côte d'Ivoire, **3** Université Angers, BIODIVAG, Angers, France, **4** REHABS International Research Laboratory, CNRS-Université de Lyon 1-Nelson Mandela University, George, South Africa, **5** African Protected Areas & Wildlife, Paris, France, **6** Office Ivoirien des Parcs et des Réserves (OIPR), Direction Nord-Est, Bouna, Côte d'Ivoire, **7** Protected Areas Management Specialist, Abidjan, Côte d'Ivoire

* PaulT.Scholte@gmail.com

## Abstract

Africa's wildlife decline has received increasing attention, yet underlying reasons have remained opaque. Using generalized additive models of 25 terrestrial and aerial counts, we present West Africa's first large herbivore population trend series alongside potential drivers. Following Comoé national park's creation in 1968, large herbivore populations increased till the mid-1980s, but subsequently declined, amplified during Côte d'Ivoire's political crisis (2002–2011) when active management ceased. Between 2010–2022, populations of roan, hartebeest and waterbuck have quasi-recovered to pre-crisis numbers. The previously dominant kob, common hippopotamus and savanna elephant have remained at c. 10% of their 1970-80s numbers, however. Grasslands declined from 15 to 2% between 1979–2020, negatively impacting kob and common hippopotamus. Since 1962, surrounding human populations and cattle inside the park increased over six-fold, yet the number of rangers only doubled. These developments have resulted in a different wildlife assemblage. Species typical of long-coarse shrub savanna - hartebeest and roan – have reached pre-crisis levels, contrary to kob and common hippopotamus likely because of the reduction of floodplain grasslands and their gregarious distribution rendering them vulnerable to poaching. We recommend increased efforts to understand habitat changes and poaching pressures, prior to re-introducing extinct species. This study highlights the importance but also the challenges of studying large herbivore populations trends alongside drivers of change.

**Data availability statement:** The Supporting Information files contain all sources and references on which the analysis builds. For further data inquiries, fellow researchers can reach out to Mr. Roger Kouadio (kouadioyaoroger2020@gmail.com), the Director of Comoe National Park.

**Funding:** The author(s) received no specific funding for this work.

**Competing interests:** The authors have declared that no competing interests exist.

## Introduction

Reports on the decline of wildlife within African protected areas have multiplied since the early 2000s [1]. Initially, major declines of large mammal populations seem to have been limited to the protected areas of West Africa, but more recently they have also been reported from Central- and East Africa [2,3,4]. Unfortunately, the drivers underlying these declines remain poorly understood, and one may only speculate on the importance of natural factors, such as declining long-term rainfall and changes in vegetation or if human pressures such as poaching or livestock intrusion cause these changes [5]. This seems to hold especially for West Africa, where because of political and institutional instability, it has been difficult to implement large mammal inventories with comparable methodology and assess potential drivers of change over decades' long periods. The lack of understanding population trends and drivers has hampered the development of targeted policy directions and management interventions [5]. Its importance was recently highlighted by an analysis of large herbivore population declines in seven protected area in Central Africa, that stressed the importance of adequate conservation inputs, based on long-term funding and political commitment [4].

Comoé National Park (Côte d'Ivoire) is with 11488 km² - the size of Gambia or Qatar - the largest protected area in the West African savannas and inscribed as World Heritage Site under criteria ix - outstanding example of ecological and biological processes - and x - outstanding biodiversity. Comoé NP has been known for its large mammal populations of kob (*Kobus kob*), lion (*Panthera leo*), African buffalo (*Syncerus caffer*), savanna elephant (*Loxodonta africana*) and common hippopotamus (*Hippopotamus amphibius*). However, a 'tremendous decrease in all mammal species between 1978 and 1998', was reported by Fischer & Linsenmair [6]. Between 2002 and 2011, the situation in Comoé National Park (NP) further degraded due to political unrest. Since 2012, the political situation improved, and in 2017 Comoé NP was "the first World Heritage site in West and Central Africa [...] to be removed from the danger list", as "species populations […] are on the rise [...] thanks to effective management of the park following a stabilization of the political situation in 2012" [7]. However, in December 2018, Comoé NP has been labelled 'red zone' by several western embassies, because of incursions of jihadists from Burkina Faso restricting especially international tourists and researchers [8]. The impact, direct or indirect, of these developments on wildlife has remained unknown so far. Because of their ecological and economic importance, large herbivore populations in Comoé NP have been assessed through guesstimates, terrestrial and increasingly aerial sample surveys since the late 1950s, only interrupted between the late 1990s and 2010, with aerial surveys continuing in 2016, 2019 and 2022. Results have been reported for selected species and limited time range only [9,10]. Moreover, no further analysis of the available time-series (1958–2022) nor of potential drivers have been made, hampering understanding of observed changes, and limiting management recommendations.

Here we analyze 60-years-long large herbivore population trends, and with the heterogeneity of count methods, opted for using generalized additive models (GAM) [3,4]. We link these trends to the potential drivers average annual rainfall and

vegetation cover, as well as surrounding human population density, number of cattle inside the park, and, as proxy for conservation efforts, the number of park guards [4,5]. We hypothesize that large herbivore populations show varying levels of resilience, depending on their vulnerability to natural factors such as declining rainfall and changing vegetation patterns as well as human pressure including livestock intrusion. We address some recommendations on future research and discuss management consequences of changes in large herbivore population composition, including rewilding initiatives. With this study we also want to stimulate protected area managers and applied scientists to team up in analyzing previous inventories of other protected areas to show the changes in large herbivore populations and the drivers that might explain them [11].

## Methods

### Study area

The area around the Comoé River was historically sparsely populated due to poor soils, river blindness disease, and high density of tsetse flies, vector for sleeping sickness. In 1926 the area between the Comoé River and Bouna was declared *Refuge Nord de la Côte d'Ivoire*, in 1953 enlarged to *Réserve de Faune et forêt classée de Bouna*, with rudimentary protection and allowing trophy hunting (S1 Table). The area west of the Comoé river (*Forêt classée de Kong*) was added in 1968 encompassing a total of 11488 km² that, with raised national park status, made it the largest savanna park in West Africa. The park is bordered in the West by two so-called biodiversity areas (faunal reserves) of 5703 km² that locally function as buffer zones (Fig 1).

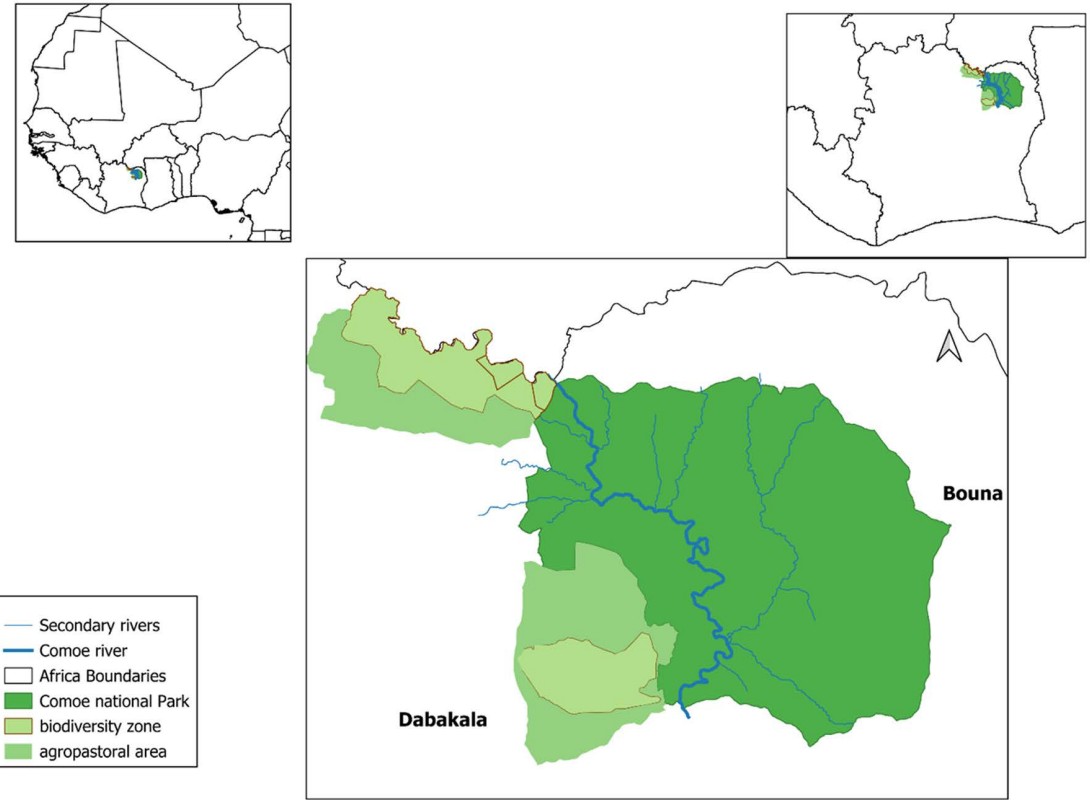

**Fig 1. Comoé national park and neighboring biodiversity buffer zones.**

Comoé NP is located in the Sudano-Guinean domain between the 1000–1200 mm isohyet, with rainfall following a unimodal rhythm. Average annual temperature is 28°C to 34°C. The weather is characterized by three seasons: a dry cool season from November to late February, a hot season from March to May, and a rainy season between June and October. Slightly undulating with elevations under 500 m, the park is crossed by the Comoé River that flows year-round from North to South, a stretch of some 240 km (Fig 1). The habitat is characterized by grassland to woodland savanna, with tree cover increasing towards the South. Shrub savannas (65% cover in 2014) are characterised by the shrubs and trees *Detarium microcarpum, Piliostigma tonningii, Crossopteryx febrifuga, Terminalia spp, Vitellaria paradoxa*, and grasses *Andropogon schirensis, A. chinensis, Schizachyrium sanguineum, Hyparrhenia subplumosa.* Tree savannas (25%) are composed by *Vitellaria paradoxa, Isoberlinia doka, Teminalia laxiflora, T. macroptera, Daniella oliveri, Lophira lanceolata, Borassus aethiopium, Burkea africana, Afzelia africana.* Forest islands (5% cover in 2014) are composed of *Vitellaria paradoxa, Crossopteryx febrifuga, Monotes kerstingii, Terminalia avicennioides and Detarium microcarpum.* Along the main rivers Comoé and Irvingo are forest galleries (4%) dominated by *Anogeissus leiocarpus, Cola cordifolia and Balanites wilsoniana.* They are in some places bordered by floodplain grasslands (2%) characterised by *Mytragina inermis, Combretum glutinosum, Daniella oliveri,* and the grasses *Andropogon africanus* and *Sporobolus pyramidalis.* Especially in the North-West, sparsely vegetated so-called Bowals with shallow lateritic soils occur (<1%), [12, 13, 14].

## Large mammal species and surveys

Comoé NP has a diverse wildlife with a total of 152 mammal species, of which 69 large mammals including nine primate species with the endemic white-naped mangabey (*Cercocebus lunulatus*) and chimpanzee (*Pan troglodytes*) as most striking ones [15,16]. Amongst the 16 carnivore species are leopard (*Panthera pardus*) and spotted hyena (*Crocuta crocuta*), as well as lion that went extinct in 2010 [9,16]. The historic presence of wild dog (*Canis pictus*) and especially cheetah (*Acinonys jubatus*) has been subject of discussion [17] (see S1 Appendix). Only recently, systematic large carnivores' counts have been conducted, not allowing long-term population trend analysis as yet. Large carnivores are therefore excluded from this analysis, see however earlier guesstimates in S1 Appendix.

From the 21 large herbivore species that occur in Comoé NP, seven have been subject of repeated surveys, subject of this study, i.e., savanna elephant, common hippopotamus, African buffalo, kob, waterbuck (*Kobus ellipsiprymnus*), hartebeest (*Alcelaphus buselaphus*) and roan (*Hippotragus equinus).* All seven species have been surveyed by direct observations through guesstimates (1958, 1968), ground surveys (1968–2012) and aerial systematic reconnaissance surveys using rear-seat observers (1977–2022), the 2022 one parallelly using an oblique camera count approach (S2 Table). We made an exception to include dung counts of savanna elephants (2016–2021), confined to forest patches in the dry season, undetectable to the aerial surveys held in that period. We considered 31 counts of large herbivores that, as far as we are aware off, have been held in Comoé NP. This includes 19 multi-large herbivore species counts, six targeting hippos, four elephants, one targeting kob and one African Buffalo. Six counts were not analyzed because of low sampling intensity (<1%), or partial cover only, rendering a park-wide extrapolation impossible, see S2 Table for more details.

For several large herbivore and primate species subject of the here considered surveys, the heterogeneity of survey methods prevents establishing reliable long-term population trends. These species are too small to be detected in aerial surveys that dominated the post-crisis period (2010–2022) i.e. oribi (*Ourebia ourebi*), common warthog (*Phacochoerus africanus*), olive baboon (*Papio anubis*), and six duiker species often lumped as 'duiker', i.e., the savanna bush duiker (*Sylvicapra grimmia*) and red-flanked duiker (*Cephalophus rufilatus*), and the forest Maxwell duiker (*Philantomba maxwelli*) and rare black duiker (*Cephalophus niger*) yellow-backed duiker (*Cephalophus sylvicultor*) and bay duiker (*Cephalophus dorsalis*). Bushbuck (*Tragelaphus scriptus*) is a rather hidden species grossly under-estimated in aerial surveys compared to terrestrial counts [18]. For reasons of documentation and transparency we present their population trends in S1 Fig. Several more elusive large herbivore species, confined to the forested parts of the park, have never been systematically surveyed, i.e., the giant forest hog (*Hylochoerus meinertzhageni*), red river hog (*Potamochoerus porcus*), water

chevrotain (*Hyaemoschus aquaticus*) and bongo (*Tragelaphus euryceros*) as well as the rare savanna species Bohor reedbuck (*Redunca redunca*) [15,19]. Doubt reigns on the occurrence of bongo and, historically, of black rhinoceros (*Diceros bicornis*) and Western giant eland (*Tragelaphus derbianus derbianus*) [15], see S1 Appendix for more details.

## Potential drivers

We obtained annual rainfall data from 1958 to 2022 through the national meteorological service for the two stations bordering Comoé NP, i.e., Bouna (NE) and Dabakala (SW), (Fig 1). We used the vegetation cover studies based on satellite imagery of 2020 (Sentinel-2, Jan-Feb), 2017 (Landsat 8, Jan-Feb), 2014 (SPOT, Dec.) and 2004 (no details available), [14,20,21]. In addition, we refer to Lauginie [19] who, based on Poilecot [13] refers to FGU Kronberg [12], who analyzed aerial photography for Comoé NP's southern part (1972: 1:40 000) and northern part (1975: 1:50 000) with ground and plane-based truthing in 1979, date that we considered here. Defining shrubs as <5m and trees as>5m, we considered grassland, bare/open land and Bowals as one category (grass & open) with a shrub and/or tree cover<5%; shrub savanna with shrub cover 5–60%; tree savanna with a tree cover 5–60%, and forests (gallery -, island- and open forests) with tree cover>60% [21]. Forest cover has been subject of discussion with an estimated 50% under reporting in 2004, when forests outside forest galleries, especially forest islands, were not distinguished from tree savanna [14].

Human demographic data were available from the country-wide censuses carried out in 1963, 1975, 1988 [22], 1998 and 2014 [16] and 2021 [23]. We considered the *sous-prefectures* (districts) directly bordering the park, using human population density figures, more comparable than absolute figures, as some of the *sous-prefectures* boundaries have changed between 1988 and 1998. We used, as proxy for financial inputs, the number of guards and other technical park personnel, lumped as 'guards' hereafter.

Cattle numbers have been reported in the post 2010 aerial surveys that have often been conducted at the end of the dry season or early rainy season when cattle are in the vicinity of protected areas and expected to have been at their highest levels. Contrary to most of the considered wild large herbivore species, the presence of cattle inside Comoé NP is seasonal for an estimated four months. Livestock biomass - but not livestock numbers - has been corrected accordingly. The 1958–1998 surveys did not report cattle number, although the presence of cattle was reported 6 km inside the park at Ouango Fetini in the North-West of the park already in 1968 [24].

As guards abandoned the park during much of the conflict years 2002–2011, their number has been set at 0 for that period [16]. 1977 data were obtained from FGU Kronberg [12], 1993–2004 from Fischer [25], 2011 from OIPR [16], and 2019 and 2022 from unpublished annual park service reports of 2019 and 2022.

## Data analysis

To accommodate the heterogeneity of the analyzed 25 counts, we modelled the change in numbers of large wild herbivores over time with generalized additive models (GAM) which have previously been successfully used to analyse wildlife trends [3,4]. We include year as a fixed factor with a negative binomial error distribution and log link function, and used a cubic B-spline covariance structure with a cubic difference penalty on the B-spline coefficients using 'gam' function in the 'mgcv' R package [26,27]. We assessed the approximate significance of smooth terms (i.e., year) with Chi-square ($\chi^2$) and its estimated degree of freedom (est.df) [27]. Confidence intervals (CI) around predicted values (indicating uncertainty) increase steeply with long bouts between consecutive surveys. We therefore checked that significant trends with time were unequivocal by plotting fitted values (±CI).

The same GAM procedure has been applied for vegetation cover (for each of the four types), metabolic biomass of wild and domestic large herbivores and number of guards. Rainfall variation (i.e., the deviation of 5-year average annual value) with time was examined using generalized least-squares (GLS) regression considering a first-order auto-regressive procedure with 'correlation=corAR1' and 'gls' function in the 'nlme' R package [28]. The variation of human population density with time was modelled with a generalized linear model (GLM) considering a log link function.

To test the effects of the potential drivers on population trends and wild & domestic herbivore biomass, we used linear regressions. When population numbers or biomass were not available from the same year as drivers, regressions were performed using predicted values from GAM. Because of the large set of drivers, it was not possible to include all drivers in a comprehensive full regression model. We therefore run separate regression to test for the effects of the eight potential drivers on wild herbivore biomass.

We performed diagnostics for all fitted generalized GAM on large mammal populations estimates and drivers. We used the 'k.check' and 'gam.check' functions to i) control that our basis dimension choices for the smooth effects were adequate, ii) get information on the convergence of the smoothness selection optimization, and iii) run diagnostic tests on the distribution of residuals and the variance of residuals against linear predictors [27]. For all other models (i.e., GLS, GLM and linear regression), we checked that residuals fulfilled statistical requirements including the distribution of residuals against fitted values and the lack of temporal autocorrelation using the 'acf' functions [28]. All analyses were performed using R 4.3.1 [29].

## Results

### Temporal trends in large herbivore biomass

After a stable period 1958-1968-1974, wild biomass increased rapidly to reach its maximum in the late 1970s – early 1980s, declining steadily till 2010, after which a remarkable recovery has taken place (GAM: $\chi^2 = 11.780$, est. df = 13.574, P = 0.030; Fig 2A). In the regular wildlife counts, livestock only appeared from 2010 onwards, constituting a fifth of year-round herbivore biomass in 2022, with broad confidence intervals however (GAM: $\chi^2 = 32.800$, est. df = 1.033, P < 0.001; Fig 2B).

### Temporal trends of large herbivore populations

The guesstimates of 1958 and 1968 are similar for most species, with numbers steadily increasing till the mid-1980s. Roan (GAM: $\chi^2 = 4.040$, est. df = 1, P = 0.044; Figure 3A) and hartebeest (GAM: $\chi^2 = 10.580$, est. df = 3.606, P = 0.048; Fig 3B) showed

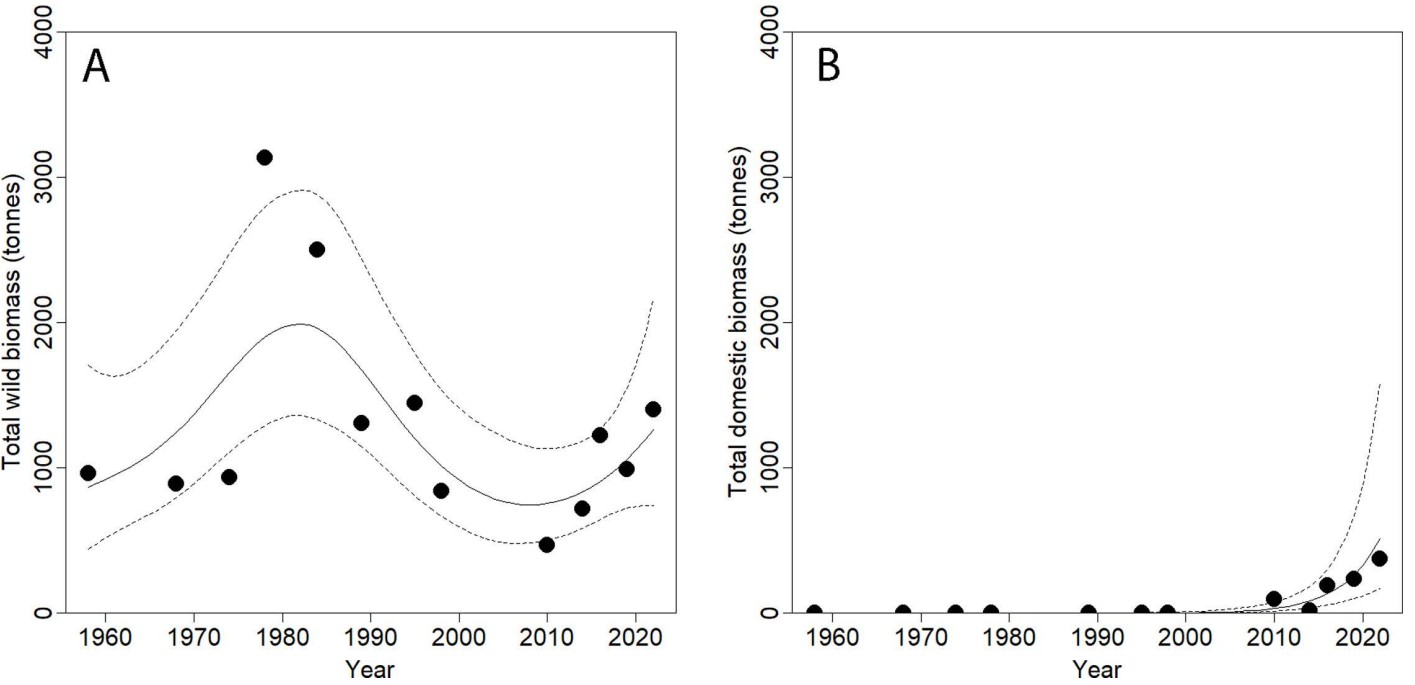

**Fig 2. Variation of wild (A) and domestic (B) metabolic biomass of large herbivores with time inside Comoé NP.**

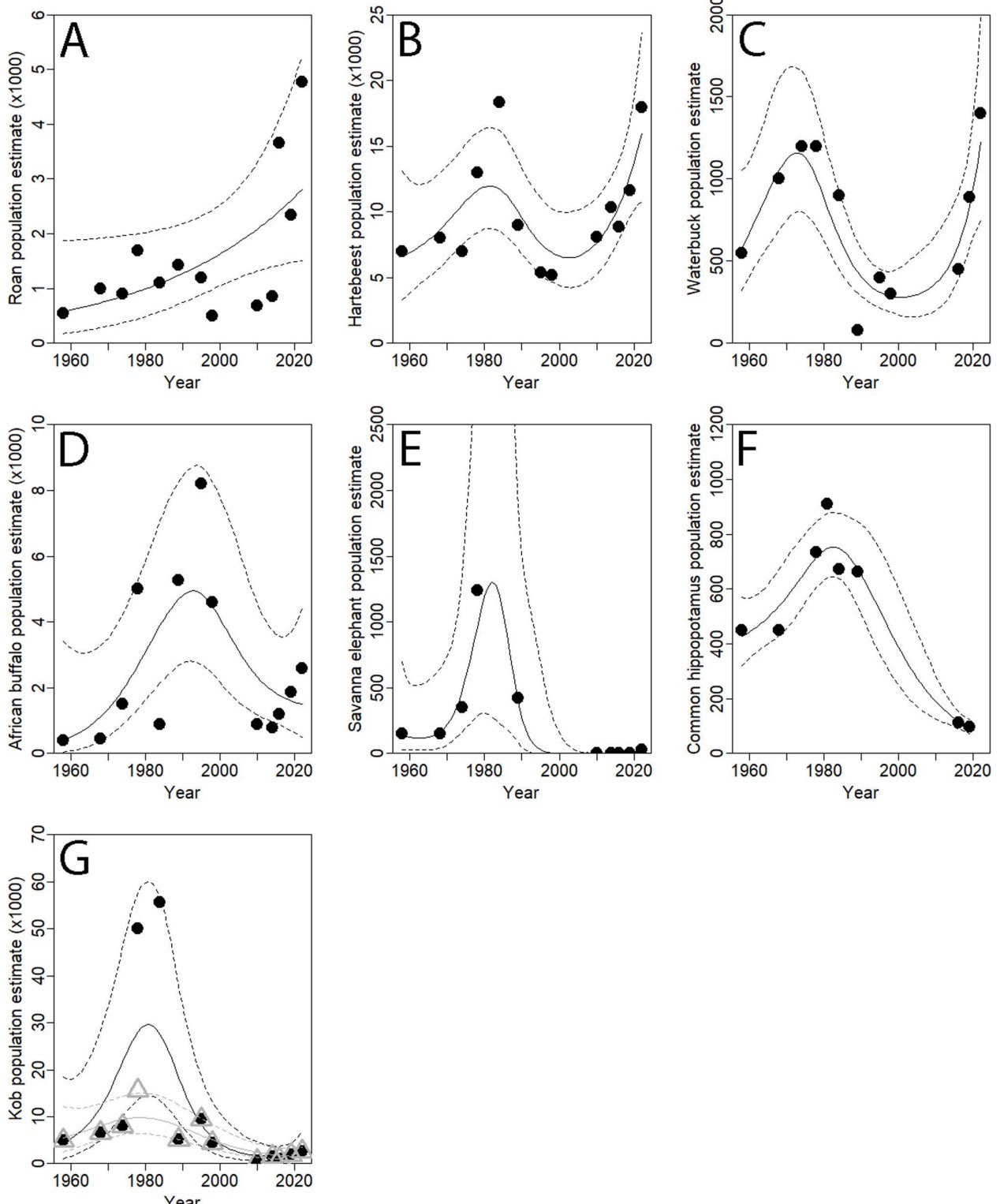

**Fig 3. Temporal trends of individual large herbivore populations, roan (A), hartebeest (B), waterbuck (C), African buffalo (D), savanna elephant (E), common hippopotamus (F) and kob (G), from 1958 to 2022.** Note two versions of kob populations numbers (V1: grey/triangles and V2: black/circles), see text.

subsequently a gradual decline till 2010, whereas waterbuck declined more suddenly (GAM: $\chi^2 = 27.990$, est. df = 3.906, P < 0.001; Fig 3C). All three species have shown a steady recovery since 2010, with roan reaching numbers seemingly double the ones of the 1970-80s. However, all three species show broad confidence intervals calling for caution to draw firm conclusions at this stage. Numbers of African buffalo have also steadily recovered but remain well below the maxima of the 1980-90s (GAM: $\chi^2 = 13.660$, est. df = 3.162, P < 0.001; Fig 3D). Like the other species the number of common hippopotamus has initially increased, reaching 900 individuals in 1970-80s, dropping dramatically and continuously to 113 and 97 in 2016 and in 2019 (GAM: $\chi^2 = 176.700$, est. df = 3.054, P < 0.001; Fig 3F). We present two GAMs for kob antelope, as widely different numbers have been reported in the mid-1980s, [19,22] in triangles and grey (V1: GAM: $\chi^2 = 16.820$, est. df = 2.532, P < 0.001; Fig 3G), and [6,12,30] in circles and black (V2: GAM: $\chi^2 = 25.380$, est. df = 3.339, P < 0.001; Fig 3G). These versions 1 and 2 show similar values except in the maxima in the late 1970 – 1980s, of c. 18 000 and 55 000 respectively; the grey version showing much tighter confidence intervals than the black version, however. Recovery of kob has been very timid, and whereas kob was the dominant large herbivore species in the 1970-1980s with the equivalent of c. 15–40% of overall wild biomass, it is only the fourth numerous species in 2022 with the equivalent of c. 4% of overall wild biomass. Savanna elephants were with c. 150 individuals relatively rare in the 1950-70s, increasing to some 400 in the 1980s (with an outlying 1240 in 1978), strongly declining afterwards and not found in the 1990s surveys (GAM: $\chi^2 = 30.910$, est. df = 3.857, P < 0.001; Fig 4E). No direct observations of savanna elephant were recorded in the aerial surveys all during the hot dry season between 2010–2016, with 2 and 28 in the aerial surveys of 2019 and 2022 respectively at the start of the rainy season. The pedestrian surveys based on dung counts in 2016–2017, 2018 and 2020, showed continuing presence of c. 60 savanna elephants in the forest patches in the South-West, also detected by camera trap imagery in 2021.

## Trends of potential drivers

In the northern part of the park, rainfall has steadily declined over the past six decades as represented by the Bouna meteorological station (GLS: F = 60.371, num df = 1, P < 0.001; Fig 4A), contrary to a curvilinear trend in the southern part, represented by Dabakala (GLS: F = 36.510, num df = 2, P < 0.001; Fig 4B).

Grass- and open land cover in Comoé NP has declined from 15% to only 2% (GAM: $\chi^2 = 22.250$, est. df = 1, P < 0.001; Fig 4C) and tree savanna has increased from 7 to 20% (GAM: $\chi^2 = 7.128$, est. df = 1, P = 0.0.008; Fig 4E) over the past 50 years with no significant variation for shrub savanna (GAM: $\chi^2 = 0.005$, est. df = 1, P = 0.942; Fig 4D) and forest cover (GAM: $\chi^2 = 0.027$, est. df = 1.806, P = 0.870; Fig 4F).

Human population densities increased 6-fold, from four (1963) to 24 people km$^{-2}$ (2021) with large regional differences: from one (West) - 12 (East) in 1963 to 12 (West) - 62 (East) people km$^{-2}$ in 2021. Expressed as total human population, human numbers increased from 84 655 to 532 216 between 1963–2021 (GLM: $\chi^2 = 28.624$, df = 1, p < 0.001; Fig 4H). Cattle number observed inside the park increased from 5806 in 2010 to 22 831 in 2022 (GAM: $\chi^2 = 14.480$, est. df = 1, P < 0.001; Fig 4G). In contrast to these increasing human pressures, the number of guards has increased from some 60 to 'only' 100 between 1977 and 2023, with guards de-facto abandoning the park between 2002 and 2011 (GAM: $\chi^2 = 7.418$, est. df = 2.675, P = 0.059; Fig 4I).

## Potential drivers and large herbivore population trends

Here we report the significant relationships between wild biomass and abundance of species and the potential drivers mentioned above, with all statistics presented in Table 1. The only significant relationship between rainfall (at Dabakala) and large herbivore populations is positive and is with the kob population (V2). Vegetation seems to have influenced several populations trends. The relationship between grass & open land and shrub savanna and kob population is positive, between grass & open land and savanna elephant population is also positive, between tree savanna and kob and elephant populations are negative whereas it is positive with roan population. The relationship between forest and waterbuck population is positive. Human population density is negatively related with kob and common hippopotamus populations,

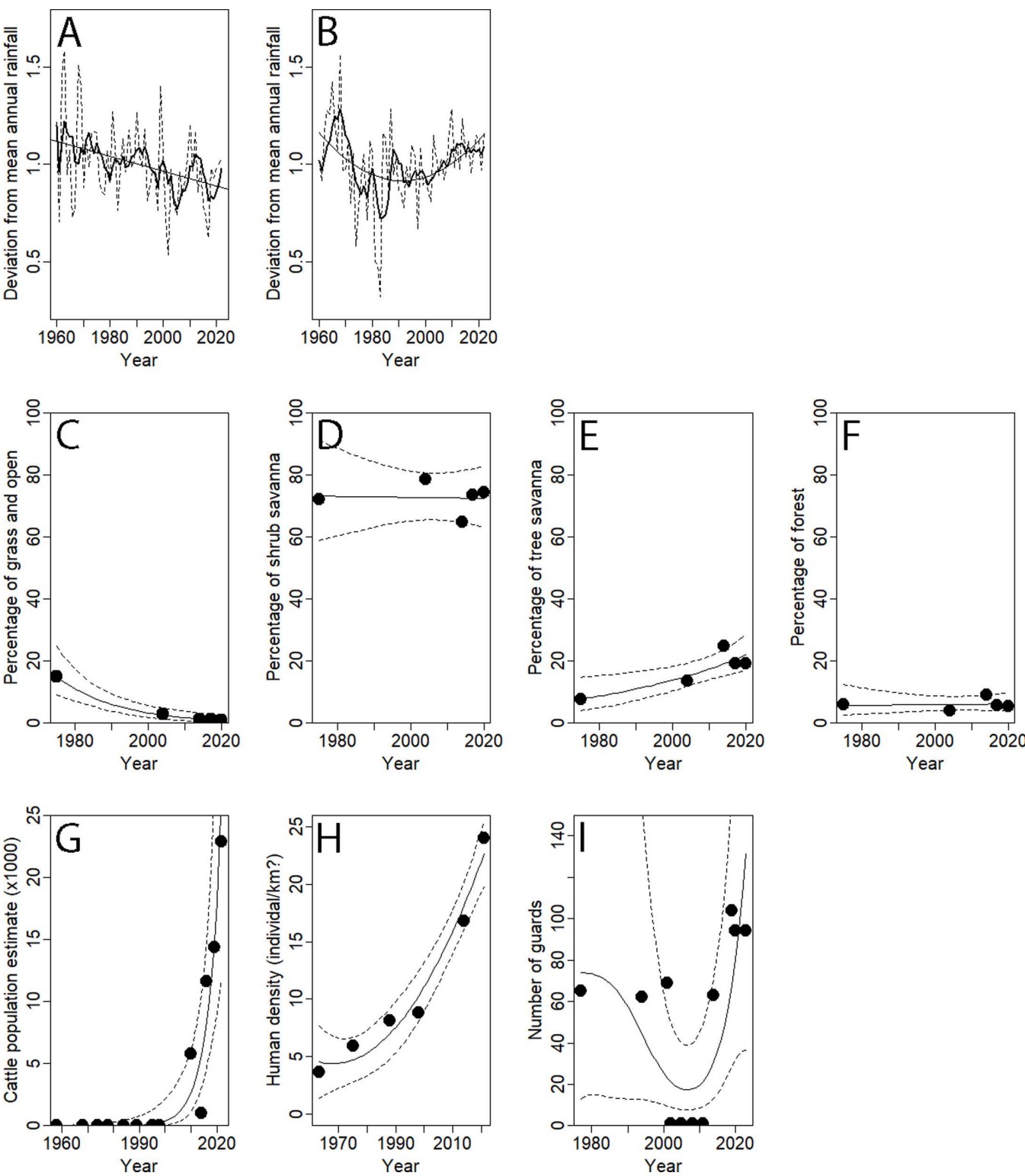

**Fig 4. Variation of potential drivers with time.** Average annual rainfall Comoé NP - North-East (Bouna 1154 mm) in **(A),** and average annual rainfall Comoé NP - South-West (Dabakala 1063mm) in **(B),** with annual average, and 5-years running average as broken and solid lines respectively. Grassland and open (includes floodplain grasslands, Bowal, see text) in **(C),** shrub savanna in **(D),** tree savanna in **(E),** Forest (includes open forest, gallery forests and forests islands) in **(F).** Number of cattle observed inside Comoé NP in **(G),** human population in the districts bordering Comoé NP in **(H),** number of rangers deployed in Comoé NP in **(I).**

**Table 1. Effects of predicted drivers on large herbivores' biomass and population numbers (see methods). β±SE indicate the estimates with the standard error of the potential drivers using linear models.**

| Predicted drivers | Large Mammal Biomass | | | |
|---|---|---|---|---|
| | B | SE | t | P |
| Rainfall Bouna | 1979.000 | 2460.000 | 0.805 | 0.438 |
| Rainfall Dabakala | 68994.000 | 85750.000 | 0.805 | 0.438 |
| Percentage of grass & open | 50.070 | 36.200 | 1.383 | 0.194 |
| Percentage of shrub savanna | 492.700 | 614.000 | 0.802 | 0.439 |
| Percentage of tree savanna | -33.780 | 35.460 | -0.953 | 0.361 |
| Percentage of forest | -774.300 | 942.100 | -0.822 | 0.429 |
| Domestic biomass | -0.742 | 1.829 | -0.406 | 0.693 |
| Human density | -31.870 | 32.880 | -0.969 | 0.353 |
| Number of guards | 11.208 | 8.157 | 1.374 | 0.197 |
| | **Roan** | | | |
| Rainfall Bouna | -5.381 | 4.248 | -1.267 | 0.231 |
| Rainfall Dabakala | -296.6 | 123.9 | -2.393 | **0.036** |
| Percentage of grass & open | -0.099 | 0.061 | -1.640 | 0.129 |
| Percentage of shrub savanna | -2.121 | 0.888 | -2.389 | **0.036** |
| Percentage of tree savanna | 0.138 | 0.048 | 2.085 | **0.016** |
| Percentage of forest | -0.045 | 0.121 | -0.374 | 0.716 |
| Domestic biomass | 0.009 | 0.001 | 6.362 | **<0.001** |
| Human density | 0.132 | 0.044 | 3.005 | **0.012** |
| Number of guards | 0.024 | 0.013 | 1.855 | 0.091 |
| | **Hartebeest** | | | |
| Rainfall Bouna | -4.050 | 14.960 | -0.271 | 0.792 |
| Rainfall Dabakala | -445.600 | 486.400 | -0.916 | 0.379 |
| Percentage of grass & open | -0.067 | 0.223 | -0.300 | 0.769 |
| Percentage of shrub savanna | -3.187 | 3.482 | -0.915 | 0.380 |
| Percentage of tree savanna | 0.210 | 0.201 | 1.044 | 0.319 |
| Percentage of forest | 0.155 | 0.402 | 0.385 | 0.707 |
| Domestic biomass | 0.016 | 0.009 | 1.824 | 0.095 |
| Human density | 0.203 | 0.186 | 1.095 | 0.296 |
| Number of guards | 0.096 | 0.041 | 2.340 | **0.039** |
| | **Waterbuck** | | | |
| Rainfall Bouna | 188.100 | 15553.200 | 0.121 | 0.906 |
| Rainfall Dabakala | 719.300 | 55532.600 | 0.013 | 0.990 |
| Percentage of grass & open | 19.370 | 24.410 | 0.794 | 0.448 |
| Percentage of shrub savanna | 6.038 | 397.598 | 0.015 | 0.988 |
| Percentage of tree savanna | 5.575 | 22.956 | 0.243 | 0.814 |
| Percentage of forest | 75.360 | 34.910 | 2.160 | **0.050** |
| Domestic biomass | 1.214 | 1.007 | 1.206 | 0.258 |
| Human density | 6.961 | 21.157 | 0.329 | 0.749 |
| Number of guards | 15.123 | 4.061 | 3.724 | **0.004** |
| | **African buffalo** | | | |
| Rainfall Bouna | -4.849 | 8.464 | -0.573 | 0.578 |
| Rainfall Dabakala | -35.800 | 288.490 | -0.124 | 0.903 |
| Percentage of grass & open | -0.030 | 0.127 | -0.242 | 0.813 |
| Percentage of shrub savanna | -0.265 | 2.065 | -0.128 | 0.900 |

*(Continued)*

**Table 1.** (Continued)

| Predicted drivers | Large Mammal Biomass | | | |
|---|---|---|---|---|
| | B | SE | t | P |
| Percentage of tree savanna | -0.029 | 0.120 | -0.238 | 0.817 |
| Percentage of forest | -0.343 | 0.207 | -1.661 | 0.125 |
| Domestic biomass | -0.003 | 0.006 | -0.596 | 0.563 |
| Human density | -0.036 | 0.111 | -0.331 | 0.756 |
| Number of guards | -0.011 | 0.028 | -0.401 | 0.696 |
| | **Savanna elephant** | | | |
| Rainfall Bouna | 744.400 | 1495.300 | 0.498 | 0.632 |
| Rainfall Dabakala | 62924.000 | 41550.000 | 1.514 | 0.168 |
| Percentage of grass & open | 36.510 | 17.010 | 2.246 | **0.054** |
| Percentage of shrub savanna | 424.100 | 226.500 | 1.872 | 0.088 |
| Percentage of tree savanna | -27.130 | 12.670 | -2.142 | **0.055** |
| Percentage of forest | 63.400 | 39.820 | 1.592 | 0.150 |
| Domestic biomass | -1.287 | 0.924 | -1.393 | 0.201 |
| Human density | -28.550 | 15.580 | -1.833 | 0.104 |
| Number of guards | 3.380 | 5.018 | 0.674 | 0.520 |
| | **Common hippopotamus** | | | |
| Rainfall Bouna | 1663.000 | 1029.000 | 1.617 | 0.157 |
| Rainfall Dabakala | 60930.000 | 39000.000 | 1.562 | 0.169 |
| Percentage of grass & open | 29.420 | 17.620 | 1.670 | 0.146 |
| Percentage of shrub savanna | 434.700 | 279.500 | 1.555 | 0.171 |
| Percentage of tree savanna | -33.49 | 14.690 | -2.280 | 0.063 |
| Percentage of forest | 4.764 | 37.125 | 0.128 | 0.902 |
| Domestic biomass | -2.522 | 0.624 | -4.040 | **0.006** |
| Human density | -32.910 | 13.160 | -2.501 | **0.046** |
| Number of guards | 4.492 | 8.114 | 0.554 | 0.600 |
| | **Kob (V1)** | | | |
| Rainfall Bouna | 19.047 | 65.279 | 0.292 | 0.776 |
| Rainfall Dabakala | 2685 | 2048 | 1.311 | 0.217 |
| Percentage of grass & open | 1.582 | 0.0854 | 1.852 | 0.091 |
| Percentage of shrub savanna | 19.190 | 14.670 | 1.308 | 0.218 |
| Percentage of tree savanna | -1.289 | 0.835 | -1.544 | 0.151 |
| Percentage of forest | 1.915 | 1.669 | 1.147 | 0.276 |
| Domestic biomass | -0.051 | 0.043 | -1.171 | 0.266 |
| Human density | -1.221 | 0.771 | -1.583 | 0.141 |
| Number of guards | 0.18 | 0.235 | 0.846 | 0.416 |
| | **Kob (V2)** | | | |
| Rainfall Bouna | 11.579 | 14.94 | 0.775 | 0.456 |
| Rainfall Dabakala | 1026.1 | 408.9 | 2.509 | **0.031** |
| Percentage of grass & open | 0.528 | 0.1627 | 3.246 | **0.008** |
| Percentage of shrub savanna | 7.337 | 2.930 | 2.504 | **0.031** |
| Percentage of tree savanna | -0.468 | 0.163 | -2.874 | **0.017** |
| Percentage of forest | 0.6047 | 0.3675 | 1.645 | 0.131 |
| Domestic biomass | -0.017 | 0.009 | -1.871 | 0.091 |
| Human density | -0.439 | 0.15 | -2.924 | **0.015** |
| Number of guards | 0.045 | 0.05 | 0.91 | 0.384 |

and positively with roan populations. This negative relationship also holds between domestic herbivore biomass and common hippopotamus whereas it is positive with roan populations. The number of guards is positively related with the hartebeest population trend.

## Discussion

### Count methodology

Large herbivores in Comoe NP have been assessed with guesstimates in 1958 and 1968, followed by systematic terrestrial and increasingly aerial counts, S2 Table. Aerial surveys have been conducted with varying intensity, from 3.3 to 20%, yet 15% seems to have been the minimum allowing proper population estimates [31]. Terrestrial counts remain a challenge in such a large protected area, and may only be useful in combination with aerial surveys, especially to allow estimates of small and hidden species, or with specific habitat requirements such as with savanna elephant. Although changes in vegetation have taken place, most notably a shift from grass & open land into denser tree savanna, we here assume that they are not sufficiently dramatic to have caused a count bias resulting in the here presented changes in population sizes.

20 out of the 25 analyzed counts have been based on direct observations, providing direct populations estimates. In 2018, the African buffalo population was assessed based on droppings, providing estimates of 2306 (rainy season) and 1715 (dry season) buffaloes, well in line with the here presented dry season aerial survey data of 2019 (1860) [32], Fig 3D. Savanna elephants form a special case, with low numbers and occurring in a very limited number of groups mostly in forest galleries or islands, easily missed in dry season aerial counts with a sample percentage of 20%. The four here considered savanna elephant dung counts confirmed their continued presence, but limited knowledge on the degradation speed of dung, as well as the large confidence intervals, should caution to interpret results as absolute population size estimations. With earlier estimates of 1240 individuals in 1978 that may be somewhat inflated or only temporary due to immigration, there is little doubt about the dramatic decline of savanna elephants with a continued existence of a vulnerable population of only c. 60 individuals in Comoé NP, the main remaining population in Côte d'Ivoire [33]. The 2022 oblique camera count, implemented in parallel to the 'classical' aerial survey, gave comparable numbers for the larger species African buffalo, hartebeest, roan and waterbuck (S2 Table). Striking are however the much higher numbers of observed kob (241%) and the here disregarded warthog (163%) [34]. Possible confusion of common non-targeted species (i.e., oribi, bushbuck, common duiker) with (young) kob, may explain some but not all difference, suggesting an underestimation of kob especially in aerial counts conducted during the late dry season, when shrubs and trees are again with leaves.

Although the recovery of roan, hartebeest and waterbuck is unmistaken, the confidence intervals of these species are considerable (Fig 3 A,B,C), highlighting the need for continued counting every 3–5 years. This should follow the same methodology of aerial surveys following the 2016 protocol, i.e., perpendicular on the watercourses and at least with moderate (15%) cover in the mid-dry season, [31]. Surveys with parallel methods, such as the 2022 aerial survey and oblique camera count, should be continued, including hitherto non-targeted species. Monitoring should also repeat the earlier (2016, 2019) total hippo dry season count along the Comoé river, if possible extended with a total kob count in the 5 km proximity of watercourses.

### Understanding large herbivore population trends

**General conservation efforts.** The relatively low numbers of large herbivores in 1958 may be attributed to the lack of conservation efforts prior to the creation of Comoé as national park, subject of Guillaume [35] who described the challenges the first park warden witnessed. With the increased protection in the years after the creation of the park and its extension in 1968, numbers increased as predicted [36]: "effective protection should result in considerable increase in

animal numbers". Except waterbuck, all species increased two-several fold between 1974 and the mid-1980s. The decline between the mid-late 1980 and 2010 was general as well, with waterbuck declining earlier and steeper than most other species. The lack of surveys between 1998–2010 prevents a more detailed assessment of the impact of Côte d'Ivoire's political crisis (2002–2011), that seems to have amplified an ongoing decline. Although the number of guards remained relatively stable (Fig 4I), management activities were said to have ceased from the late 1980s, with the main safari lodge closing in 1992 [6,25] (S2 Table). The recovery of roan, hartebeest and waterbuck, and to a lesser extend African buffalo, from 2010 onwards shows the impact of improved conservation efforts, but also these species' resilience. The recovery of hartebeest seems to be in line with the 'healthy population structure' as recently suggested [37]. We caution however for too much optimism at this stage, given the wide confidence intervals. Kob, common hippopotamus and savanna elephant have remained at c. 10% of their pre-war numbers, requiring further investigations in relation with selective predation, poaching and changes in vegetation [5], explored below.

**Selective predation.** One may exclude predation to be a main factor inhibiting the recovery of the megaherbivores savanna elephant and common hippopotamus [38]. Increased predation by meso-carnivores (especially olive baboon *Papio anubis*) could potentially explain the timid recovery of kob. Local extinction of large carnivores (lion, leopard, spotted hyena) has led in neighboring Ghana to exploding numbers of olive baboon and subsequent reduction of meso-herbivores [39]. However, olive baboon numbers in Comoé NP, albeit grossly underestimated because of the aerial surveys see Methods, do not reflect such possible explosion (S1 Fig.). Also, the capture rate of olive baboon in the 2022 camera trap survey was only 2.5%, lower than 3.8 and 3.1% of respectively tantalus monkey *Chlorocebus tantalus* and patas monkey *Erythrocebus patas* [40]. We therefore draw the cautious conclusion that olive baboon numbers have not exploded since the extinction of lion, possibly because of the continued presence of leopard [40], a known baboon predator in Central Africa [41], see also S 1 Appendix.

**Selective poaching.** Kob and common hippopotamus are highly gregarious, sensitive for disturbance by fishermen and gold delvers along the Comoé River. Both species may therefore be prime subject of poaching as dramatically shown for kob in the northern Central African Republic, being the most vulnerable species declining rapidly and with limited capacity for recovery under continued pressure [42]. Common hippopotamus has shown to be intensively poached in areas not sufficiently protected, that should come not as a surprise with the 230 km long Comoé River, resembling their decline along the Benoué River in North Cameroon [43]. Only few signs of poaching of hippopotamus (carcasses, meat drying installations, etc.) have been found along the Comoé River however. Poaching may also continue to impact savanna elephants, whose extended wandering outside the park boundaries, makes effective protection particularly difficult for this species under considerable pressure [33].

**Vegetation – large mammal relations.** We may attribute the timid recovery of kob and the continued decline of common hippopotamus to the greatly reduced cover, from 15 to 2% between 1979 and 2020, of floodplain grasslands and Bowal rainy season grounds that may have functioned as grazing lawns. We hypothesize that the decline of common hippopotamus and kob due to poaching in the late 1990s and early 2000s, has caused the disappearance of grazing lawns. In the comparable Central African dystrophic savanna, common hippopotamus has been observed as instrumental for the maintenance of grazing lawns, facilitating mesoherbivores such as kob [44, 45, 46]. However once disappeared, grazing lawns are notoriously difficult to recover especially in nutrient poor savannas such as Comoé, although under controlled circumstances fertilization has shown to be a viable method of increasing mineral levels in the soil and grass leaf [45,47].

The reduction of grasslands and parallel increase of tree savanna, are in line with the popular view amongst communities that Comoé NP is becoming increasingly wooded [14] (Fig 4E). One is tempted to link the tree savanna increase to the remaining low numbers of savanna elephants, a relation found across Africa [e.g., 48, 49, 50]. In addition, the rebounding average annual rainfall over the past decades, especially in the South may have stimulated the increase in tree cover.

## Challenges in analyzing potential drivers

Proximate drivers should be distinguished from underlying drivers, not causing change themselves, but acting indirectly and often dependent on other factors [5]. Based on earlier experiences we assessed the underlying drivers of large herbivore population trends, human population density and conservation financing, as well as the potential proximate drivers average annual rainfall, vegetation cover and number of cattle [4]. Apart from kob, and common hippopotamus, it remains difficult to link these drivers of change with large herbivore population trends. We attribute this to the overriding influence of the lack of protection in the 1990s and especially between 2002–2010. The lack of counts between 1998 and 2010, and the poor documentation of poaching, has prevented an analysis of these changes, however.

Human population in park surroundings is potentially one of the major underlying drivers of change, although culture and tradition towards conservation may play an even bigger role [5]. Although illegal, livestock, predominantly cattle, are entering especially the northern part of the park, potentially causing avoidance of wild large herbivores or competing with them [4,51]. Recurrent droughts seem to have pushed cattle herders from the Sahelo-Sudanian savanna south towards Comoé NP; this southward movement has also been favored by the decrease of trypanosomes in the study area [16]. We hypothesize that the considerable size of Comoé may have buffered, at least till now, influences such as the rapidly increasing human population in the park surroundings, and the related rapidly increasing number of cattle that seem to have remained limited to the northern edges of the park so far. The mechanisms behind these phenomena remain poorly understood and contradict conclusions drawn from mostly medium-sized protected areas in West-Central Africa where size of protected areas was not correlated with the extinction of large carnivores [17].

Long-term data on governmentally managed parks' budgets are notoriously difficult to obtain with different budget lines and origins, i.e., salaries, operational costs, investments, in addition to budget lines under other ministries, e.g., infrastructure. The number of guards suggests a steady increase, but apart from the various spells of insecurity during 2002–2011, does not take into account periods during which their effective deployment in the field might have been more limited. The number of guards was a strong driver in one of seven parks in the Central African savannas, showing that with long-term sufficient inputs, wildlife was thriving [4].

Rainfall is one of the main potentially proximate drivers, influencing primary production, forage base for large herbivores [5]. With the perspective of climate change, temperature is expected to play an increasingly important role. However, at the temporal (65 years) and spatial scale (11 000 km$^2$) of this study, analyzing the impact of temperature seems to be out of reach, with the only historical data available from outside the park.

Contrary to eastern and southern Africa, vegetation cover has seldom been considered a driver of large herbivore population trends in West-Central Africa, possibly because of poorly known large herbivore – vegetation dynamics [5,48]. This may be one of the reasons of the low frequency of vegetation surveys, with only one vegetation survey conducted in Comoé NP prior to 2004. Also, for the potential drivers, annual rainfall surveys have been few and heterogenous and it took us several years to obtain this data.

In addition to the need of standardization of counts mentioned above [31], efforts should be undertaken to harmonize data on potential drivers, often scattered in time and incomplete. Attention should also be paid to drivers of political changes and poaching as they may lead to abrupt changes in large mammal populations. This will allow running full statistical models with all potential drivers and their interactions, limiting overestimations when testing drivers on variation of large mammal populations, and clarifying mechanisms when drivers interact, e.g., habitat use and cattle-wildlife interactions [cf 51]. In addition to GAM, it would be recommendable to use other statistical procedures to model long-term wildlife trends as, for instance, Bayesian models and examine their goodness of fit [11].

## Rewilding initiatives and potential of reintroduction of lion

The decline and recovery of large mammal populations has been studied in a few other protected areas across Africa. In West-Central Africa, Waza NP (Cameroon) has been impacted by the construction of an upstream dam, causing the

collapse of floodplain species kob and korrigum (*Damaliscus lunatus*), and increase of savanna elephants. With subsequent flood releases and increase in floodplain vegetation, kob initially recovered before dropping again, likely due to poaching [4,52,53]. Zakouma (Chad) has been subject of post-war recovery with an order of magnitude increase in savanna elephant and African buffalo, the former collapsing because of poaching [4,54]. In post-war Gorongosa NP (Mozambique), waterbuck increased multifold whereas elephant and other populations remained at low levels [55]. Although these protected areas share these same large mammal species, their population development during the phases of collapse and recovery differ remarkably, and remain with the exception of elephant poaching, largely unexplained. With an increasing number of surveyed populations, cross-continent comparative studies may help further understanding these pathways, and advise rewilding initiatives on measures to be taken.

This holds in particular for the local extinction of lion and their impact on large herbivore populations. Trophic rewilding of Comoé, i.e., the reintroduction of lion (and possibly Western giant eland (S1 Appendix)) has captivated researchers and park managers over the past years [9,40]. During the 1970-80s, at the height of wildlife abundance in particular kob, lion has been reported at densities of c. 1 lion 100 km$^{-2}$, in line with observations from elsewhere in West Africa, most notably the 1.5 lions 100 km$^{-2}$ in its stronghold Pendjari – W [56]. One may question therefore the conclusion of Funston et al. [57] that in Comoé NP "[…] most of the important potential prey species for lions have recovered to some extent" and "based on these […] estimates of ungulates, the park could support about 450 lions at a density of about 3.9 lions 100km$^{-2}$".

The here presented asymmetric changes in quasi-recovered large mammal populations, raise questions to be addressed before actual lion reintroduction could take place. This holds in particular for the timid recovery of kob compared to the other large herbivores. Kob has been the main prey for lion in the Central African parks of Manovo-Gounda-St.Floris [41], Waza [58], and Faro [59], although given its abundance not necessarily the preferred one [60,61]. However, the only study on lion preys in Comoé NP concluded "the lions … preyed predominantly upon small to medium-sized ungulates" in particular kob, "while large ungulates (hartebeest, buffalo and warthog) made up a surprisingly small proportion of their diet and were not preferentially killed as recorded for eastern and southern Africa" [62].

### Large herbivores as monitors for protected areas

World Heritage site Comoé NP has outstanding biodiversity, yet despite decades of ecological research enabled by the presence of a well-equipped research station (Table S2), the here presented population trends of large herbivores are the only decades' long population trends know for any wildlife in Comoé NP. Some qualitative comparison has been made based on vulture observations between 1968–1972 and 1980–1984 [63]. The increase in wild and domestic herbivore biomass, does not seem to be matched by an increasing vulture population, suggesting other factors then availability of food might have limited the size of vulture populations, e.g., the rapidly and widespread increasing effects of poisoning [64].

### Conclusions and recommendations

Using generalized additive models (GAM), associated with extrapolation for missing data, allowed juxtaposing counts with widely varying methodology to assess and test for temporal trends in population surveys. This should not distract from the necessity to continue surveying with a standardized methodology, as with the broad confidence intervals, obtaining more precision in large mammal populations trends would be important. This study builds on proper knowledge of the main drivers of change. To understand their development in time required insight into the history of the protected area, including the changes in human populations in their surroundings, as well as from other drivers such as annual rainfall and number of guards. Regular studies on changes in vegetation and land use will also be essential for further assessments.

Over the past decades Comoé NP has passed through serious challenges such as civil war (2002–2011) with years of neglect in surveillance, declining rainfall in its northern parts, and an overall five-fold increasing human population. Nevertheless, Comoé NP once again holds important large mammal populations, albeit less diverse because of previous extinctions - most notably lion - and in a different composition as previously, with common hippopotamus, kob and

savanna elephant in much lower numbers. The enormous size of the park and improving conservation efforts are amongst the factors that may explain the partial recovery, that contrasts the dramatic decline of wildlife in Côte d'Ivoire outside protected areas, as reported in the 2019–2021 national forest and wildlife inventory [65]. We recommend only after understanding the limited recovery of kob and hippopotamus, to implement any possible reintroduction of lion [66]. Grazing lawn restoration, through intensified grazing, fire-management, and possibly fertilization should first be piloted, for which a precipitated reintroduction of lion will complicate operations.

More in general we propose concentrating scarce human and financial resources to the protection of Comoé NP's unique species and ecosystems, including chimpanzee and white-naped mangabey, as well as its diverse bird, herpeto- and insect fauna [16]. The unique characteristics of Comoé NP, being amongst the last remaining savanna wilderness areas in West-Central Africa, should be further exploited and we recommend investing in rendering them interesting for international and nationally oriented ecotourism [67].

## Supporting information

**S1 Alternative Language Abstract.  French translation of the abstract.**
(DOCX)

**S1 Appendix.  Local extinctions and status of large carnivores.**
(DOCX)

**S1 Fig.  Trends of large mammal populations, bushbuck (A), oribi (B), duikers (C), common warthog (D) and olive baboon (E) from 1958–2022.**
(TIF)

**S1 Table.  Years of major events in Comoé NP.**
(DOCX)

**S2 Table.  Data description of large herbivore counts Comoé NP (1958–2022).**
(DOCX)

## Acknowledgments

We would like to acknowledge the efforts of all counting teams. A special mention of our deceased teacher and colleagues Chris Geerling, Francis Lauginie and Pierre Poilecot, who initiated the here reported large herbivore counts, and passed away before we could discuss their beloved Comoé. We also acknowledge the pilots-aerial survey experts, Philippe Bouché and Petr Viljoen, driving forces behind the counts of 2016 and 2019, who died in later aerial missions. We acknowledge the constructive comments of two reviewers and editor. The opinions expressed in this article are solely those of the authors.

## Author contributions

**Conceptualization:** Paul Scholte, Olivier Pays, Bertrand Chardonnet, Djafarou Tiomoko.

**Data curation:** Paul Scholte, Amara Ouattara, Djafarou Tiomoko.

**Formal analysis:** Paul Scholte, Olivier Pays.

**Investigation:** Paul Scholte, Olivier Pays, Bertrand Chardonnet, Amara Ouattara, Djafarou Tiomoko.

**Methodology:** Paul Scholte, Bertrand Chardonnet, Amara Ouattara.

**Project administration:** Paul Scholte.

**Supervision:** Paul Scholte.

**Validation:** Paul Scholte, Bertrand Chardonnet, Djafarou Tiomoko.

**Visualization:** Paul Scholte.

**Writing – original draft:** Paul Scholte, Olivier Pays.

**Writing – review & editing:** Paul Scholte, Olivier Pays, Bertrand Chardonnet, Amara Ouattara, Djafarou Tiomoko.

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
