## [Decision Letter · Decision Letter 0]

10 Dec 2024

PONE-D-24-38244Large mammal population trends in Comoé National Park (1958-2022): towards understanding their asymmetric decline and recovery in West Africa’s largest savanna park.PLOS ONE

Dear Dr. Scholte,

Thank you for your patience with the unusually long review time for your submission to PLOS ONE. After careful consideration, we feel that it has merit but does not fully meet PLOS ONE’s publication criteria as it currently stands. Therefore, we invite you to submit a revised version of the manuscript that addresses the points raised during the review process.

We look forward to receiving your revised manuscript.

Kind regards,

Stephanie S. Romanach, Ph.D.

Academic Editor

PLOS ONE

Journal Requirements:

2. We note that Figure 1 in your submission contain map images which may be copyrighted. All PLOS content is published under the Creative Commons Attribution License (CC BY 4.0), which means that the manuscript, images, and Supporting Information files will be freely available online, and any third party is permitted to access, download, copy, distribute, and use these materials in any way, even commercially, with proper attribution. For these reasons, we cannot publish previously copyrighted maps or satellite images created using proprietary data, such as Google software (Google Maps, Street View, and Earth). For more information, see our copyright guidelines: http://journals.plos.org/plosone/s/licenses-and-copyright.

1) You may seek permission from the original copyright holder of Figure 1 to publish the content specifically under the CC BY 4.0 license.  

2) If you are unable to obtain permission from the original copyright holder to publish these figures under the CC BY 4.0 license or if the copyright holder’s requirements are incompatible with the CC BY 4.0 license, please either i) remove the figure or ii) supply a replacement figure that complies with the CC BY 4.0 license. Please check copyright information on all replacement figures and update the figure caption with source information. If applicable, please specify in the figure caption text when a figure is similar but not identical to the original image and is therefore for illustrative purposes only.

**Additional Editor Comments:**

Your manuscript represents a tremendous endeavor to highlight population trends and drivers in a part of the world that has been challenging to study. The reviewers and I agree that your manuscript should make a valuable contribution to the literature but would benefit from a few considerations and revisions. Below you will find reviews from two reviewers with expertise in large mammal conservation in savanna Africa and one with particular expertise in statistics and data analysis. Both reviewers provide helpful suggestions for the context of your findings, as well as some analytical suggestions from Reviewer 2.

Reviewers' comments:

Reviewer's Responses to Questions

**Comments to the Author**

1. Is the manuscript technically sound, and do the data support the conclusions?

Reviewer #1: Partly

Reviewer #2: Yes

2. Has the statistical analysis been performed appropriately and rigorously? 

Reviewer #1: Yes

Reviewer #2: Yes

3. Have the authors made all data underlying the findings in their manuscript fully available?

Reviewer #1: Yes

Reviewer #2: Yes

4. Is the manuscript presented in an intelligible fashion and written in standard English?

Reviewer #1: Yes

Reviewer #2: Yes

5. Review Comments to the Author

Reviewer #1: The manuscript describes interesting wildlife recovery patterns in a protected area of extreme regional importance. While the datasets used by the authors to describe and interrogate the observed wildlife trends are of very variable nature and quality, as the authors openly acknowledge, the authors appear to have done their best to accommodate the obvious flaws and variability in those datasets, by using GAMs and by interpreting the results with the required caveats.

I would therefore generally recommend to accept this manuscript for publication. However, I would strongly recommend to further strengthen the discussion by relating the observed wildlife recovery trends for the various species to common ecological concepts described for African savanna herbivore communities, which could help to explain the observed patterns. The possible relationship between low elephant numbers and a decrease in open habitats is mentioned in once sentence (lines 374-376), however, no references are given and the effect this could have on other herbivore species is not discussed. There is strong evidence for the importance of such interactions, however (e.g. de Boer et al, 2015), and this could be discussed in more detail. The situation of hippopotamus in the park is discussed in some detail, but the authors do not evaluate the ecological role of the species, and any effects their population collapse may have on other species. Like elephants, common hippo are considered to be an ecosystem engineer (e.g. Voysey et al, 2023), and their facilitative role for smaller herbivores has also been documented for Western kob (e.g. Verweij et al, 2006). This could be discussed in more detail. Furthermore, it has been well-documented in neighbouring Ghana that the local extirpation of lions (and leopards) led to dramatic trophic cascades, with meso-predators like baboons increasing their numbers 4-5-fold, which in turn led to a dramatic decline in numbers of medium-sized ungulates (Brashares et al, 2010). The local extinction of lions is mentioned several times by the authors, but no possible link to trophic cascades is explored. Baboon predation of herbivores particularly targets fawns of species like kob that hide fawns in tall grasses at the edge of grazing lawns, conversely to larger plains herbivores like hartebeest, or also the roan, for which even newborns follow herds within hours and are not overly susceptible to baboon predation. This potential role of meso-predators, which may have become hyperabundant following the local extirpation of top predators such as lions, is not discussed at all.

On the other hand, and also regarding lions, the links the authors are trying to establish between low kob numbers and potential lion reintroduction plans seem construed. Kobs are neither within the preferred prey range of lions, which is in the range of 190–550 kg (Hayward & Kerley (2005), nor was kob to be found an important prey species in dietary analyses across West & Central Africa (Bauer et al, 2008). As can be seen in Bauer et al (2008), larger-bodied species such as hartebeest and roan are generally much more important.

Lastly, it would also be beneficial to the readers if the authors discussed their findings in the context of recovery trends and patterns from other African protected areas. Some really interesting works on this matter come from Gorongosa NP in Mozambique, for example, where contemporary herbivore community structure and biomass also differs radically from the previous documented baselines (e.g. Pansu et al, 2019). There are other, similar examples from different systems, and the manuscript would really benefit from a discussion showing that such skewed recovery favouring only certain species appears to be a phenomenon which is more common than may have been anticipated, and may require specific research and management interventions to restore pre-depletion faunal assemblages.

Specific literature to consider in the discussion:

Bauer, H., Vanherle, N., Di Silvestre, I., & De Iongh, H. H. (2008). Lion–prey relations in West and Central Africa. Mammalian Biology, 73(1), 70-73.

Brashares, J. S., Prugh, L. R., Stoner, C. J., & Epps, C. W. (2010). Ecological and conservation implications of mesopredator release. Trophic cascades: predators, prey, and the changing dynamics of nature, 221-240.

de Boer, W. F., Van Oort, J. W., Grover, M., & Peel, M. J. (2015). Elephant-mediated habitat modifications and changes in herbivore species assemblages in Sabi Sand, South Africa. European Journal of Wildlife Research, 61, 491-503.

Fritz, H., Duncan, P., Gordon, I. J., & Illius, A. W. (2002). Megaherbivores influence trophic guilds structure in African ungulate communities. Oecologia, 131, 620-625.

Hayward, M. W., & Kerley, G. I. (2005). Prey preferences of the lion (Panthera leo). Journal of zoology, 267(3), 309-322.

Pansu, J., Guyton, J. A., Potter, A. B., Atkins, J. L., Daskin, J. H., Wursten, B., ... & Pringle, R. M. (2019). Trophic ecology of large herbivores in a reassembling African ecosystem. Journal of Ecology, 107(3), 1355-1376.

Verweij, R., Verrelst, J., Loth, P. E., MA Heitkönig, I., & MH Brunsting, A. (2006). Grazing lawns contribute to the subsistence of mesoherbivores on dystrophic savannas. Oikos, 114(1), 108-116.

Voysey, M. D., de Bruyn, P. N., & Davies, A. B. (2023). Are hippos Africa's most influential megaherbivore? A review of ecosystem engineering by the semi‐aquatic common hippopotamus. Biological Reviews, 98(5), 1509-1529.

Reviewer #2: Trends in large herbivore populations in Comoé National Park were investigated in relation to various drivers through a data set spanning more than six decades. Large herbivore populations are claimed to have shown a pattern of asymmetric recovery and decline based on environmental changes, poaching and management strategies during periods of political instability. The study further claims that different species are differently affected by the described drivers with some species exhibiting higher levels of recovery than others.

Against the backdrop of ecological management in West Africa, where long-term data are often found to be scarce, these claims are significant for conservational efforts and provides a rare insight into the relationship and between human activity, habitat changes, and wildlife trends which could potentially lead to changes and developments in the spheres of conservational policies.

The study is well placed within the context of existing literature, acknowledging the comparable studies from Central and East Africa, as well as similar studies investigating drivers of wildlife decline. Referencing older works are expected in a long-term wildlife study and relevant newer studies were introduced to show the use of more recent statistical techniques. One technique that could be considered to enhance the discussion, would be Bayesian approaches.

Overall, the authors have managed to effectively describe the knowledge gap that their research addresses.

The data analyses were found to largely support the claims of this manuscript and generalized additive models (GAMs) are appropriately applied in this non-linear trend environment for heterogeneous data.

Data is obtained from various sources and the authors have done well to acknowledge inconsistent methodologies and justifications for the inclusion of such data. The rarity of long-term data is an acceptable rationale, and this study seems to be the first of its kind for this region, potentially providing important and necessary insights into population trends. For future studies, one would hope to see greater effort to harmonize data for improved reliability.

The methodology is sufficiently described for reproduction, and the manuscript is generally well-organized. The manuscript is a valuable contribution to conservation science and given the suggested adjustments/addressing the highlighted concerns below, the study has merit for eventual publication.

We are of the opinion that the following aspects warrant attention:

1. The heterogeneity of the data, while acknowledged, introduces uncertainty in the results, especially where less consistently surveyed species are concerned. Inconsistent sampling techniques may limit the comparability of population estimates over time.

2. Potential interaction effects of combining multiple drivers should be investigated for deeper insights. The authors note that models incorporating all drivers are not suitable, the importance of individual variables might be overestimated.

3. Cubic B-spline covariance structure and standard assumptions of GAMs might require further investigation for data on ecological trends where instances of political crises could potentially lead to abrupt changes.

4. Confidence intervals are broad and should be interpreted with caution.

5. Graphical displays, such as Figure 2 B is unclear. Although it is acknowledged that the authors might have been aiming to keep the y-axis indices the same for both graphs A and B, we believe that it might be more sensible to shrink the indices of the y-axis of graph B to a more sensible ceiling of 2000.

6. Aligned with the requirement of the journal, we would have liked to a see a tabular summary such as S3 Table incorporated into the manuscript itself and not merely included in the appendix.

7. As a point of interest, one wonders what the effect of temperature could be on population trends. It is noted that average temperature is between 28 and 34 degrees, but would this variable not also be a driver against the backdrop of climate change?

8. Although a detailed account of the sources and methods used for data analyses is supplied, depositing the raw data in an accessible repository would enhance transparency. Furthermore, simplifying key findings with summary tables could enhance readability.

6. PLOS authors have the option to publish the peer review history of their article (what does this mean? ). If published, this will include your full peer review and any attached files.

**Do you want your identity to be public for this peer review?** For information about this choice, including consent withdrawal, please see our Privacy Policy .

Reviewer #1: No

Reviewer #2: No

---

## [Author Response · Author response to Decision Letter 1]

11 Feb 2025

Pls see the attached file (allowing easier reading)

---

## [Editor Report · Decision Letter 1]

19 Feb 2025

Large mammal population trends in Comoé National Park (1958-2022): towards understanding their asymmetric decline and recovery in West Africa’s largest savanna park.

PONE-D-24-38244R1

Dear Dr. Scholte,

We’re pleased to inform you that your manuscript has been judged scientifically suitable for publication and will be formally accepted for publication once it meets all outstanding technical requirements.

Kind regards,

Stephanie S. Romanach, Ph.D.

Academic Editor

PLOS ONE

---

## [Editor Report · Acceptance letter]

PONE-D-24-38244R1

PLOS ONE

Dear Dr. Scholte,

I'm pleased to inform you that your manuscript has been deemed suitable for publication in PLOS ONE. Congratulations! Your manuscript is now being handed over to our production team.

Kind regards,

on behalf of

Dr. Stephanie S. Romanach

Academic Editor

PLOS ONE